# Lymph Node Number Predicts the Efficacy of Adjuvant Chemoradiotherapy in Node-Positive Endometrial Cancer Patients

**DOI:** 10.3390/diagnostics10060373

**Published:** 2020-06-04

**Authors:** Jie Lee, Tsung Yu, Mu-Hung Tsai

**Affiliations:** 1Department of Radiation Oncology, MacKay Memorial Hospital, Taipei 104, Taiwan; sinus.5706@mmh.org.tw; 2Department of Medicine, MacKay Medical College, New Taipei City 252, Taiwan; 3Department of Public Health, College of Medicine, National Cheng Kung University, Tainan 701, Taiwan; tsungyu@mail.ncku.edu.tw; 4Department of Radiation Oncology, National Cheng Kung University Hospital, College of Medicine, National Cheng Kung University, Tainan 704, Taiwan; 5Department of Computer Science and Information Engineering, College of Electrical Engineering and Computer Science, National Cheng Kung University, Tainan 701, Taiwan

**Keywords:** endometrial cancer, radiotherapy, chemotherapy, lymph node

## Abstract

This study aimed to evaluate the value of lymph node (LN) number as a predictor for adjuvant treatment in node-positive endometrial cancer. Data of 441 patients diagnosed with International Federation of Gynaecology and Obstetrics (FIGO) stage IIIC endometrial cancer and who underwent adjuvant chemotherapy alone or chemoradiotherapy between 2009 and 2015 from the Taiwan Cancer Registry were reviewed. The patients were stratified based on the number of positive LN as follows: 1, 2–5, and ≥ 6. The overall survival (OS) was analysed using the Kaplan–Meier method and the Cox proportional hazards model. In multivariable analysis, chemoradiotherapy was independently associated with improved OS (hazard ratio [HR]: 0.62, 95% confidence interval [CI]: 0.43–0.90; *p* = 0.01) compared with chemotherapy alone. Patients with ≥ 6 positive LNs were associated with a worse OS (HR: 2.22, 95% CI: 1.25–3.95; *p* = 0.006) and those with 2–5 LNs were not associated with a worse OS (HR: 1.56, 95% CI: 0.94–2.59; *p* = 0.09) compared to patients with one LN. When stratified based on LN number, chemoradiotherapy was found to significantly improve the 5-year OS of patients with ≥ 6 positive LNs compared to chemotherapy alone (35.9% vs. 70.0%, *p* < 0.001). No significant differences between chemotherapy alone and chemoradiotherapy were observed in 5-year OS among patients with one LN (73.1% vs. 80.8%, *p* = 0.31) or 2–5 positive LNs (71.4% vs. 75.7%, *p* = 0.68). Lymph node number may be used to identify node-positive endometrial cancer patients who are likely to have improved OS with intensification of adjuvant therapy.

## 1. Introduction

Endometrial cancer is the leading gynaecologic malignancy in developed countries, and the mortality rate has increased by 1.9% annually between 2010 and 2015 [1,2]. Most women diagnosed with endometrial cancer are patients with early-stage disease and a favourable prognosis. However, women with International Federation of Gynaecology and Obstetrics (FIGO) stage IIIC endometrial cancer (pelvic or para-aortic lymph node involvement) are at higher risk of recurrences and cancer-related deaths [3,4,5,6].

The primary treatment for endometrial cancer is surgery, including total hysterectomy, bilateral salpingo-oophorectomy, and lymphadenectomy or sentinel lymph node (LN) mapping. Adjuvant treatments are mainly recommended based on pathological evaluation. However, the optimal adjuvant treatment for patients with node-positive endometrial cancer is controversial [5,6,7,8,9]. The adjuvant chemotherapy or radiotherapy achieved similar overall survival (OS) outcomes; however, their pattern of recurrences differed [10,11]. Combined chemotherapy and radiotherapy might achieve better survival outcomes compared with chemotherapy or radiotherapy alone in patients with node-positive endometrial cancer [11,12,13,14]. In the PORTEC-3 trial, 170 (25.8%) patients had FIGO stage IIIC endometrial cancer and patients with FIGO stage IIIC disease had high risks of recurrences. The PORTEC-3 trial demonstrated that chemoradiotherapy over radiotherapy alone improves the OS outcome in overall stage III endometrial cancers [6]. In the GOG 258 trial, 538 (73.1%) patients had FIGO stage IIIC endometrial cancer; however, subgroup analysis showed that these patients might not have benefitted more from chemoradiotherapy than from chemotherapy alone in recurrence-free survival [7]. Hence, there may be a need to select patients with FIGO stage IIIC endometrial cancer to receive chemoradiotherapy or chemotherapy alone.

Increasing number of LN involvements is associated with worse OS outcomes in patients with node-positive endometrial cancer [4]. Thus, we hypothesised that nodal number could be an indicator in deciding whether patients with node-positive endometrial cancer would benefit from chemoradiotherapy or chemotherapy alone. However, no prospective randomised trials have evaluated this approach, and it is yet unknown which group of patients would benefit from chemoradiotherapy and which from chemotherapy alone. Therefore, this study aimed to investigate the nodal number as a predictor for guiding adjuvant treatment in patients with node-positive endometrial cancer using the national dataset.

## 2. Materials and Methods

### 2.1. Database

All data in this study were retrieved and analysed within the Health and Welfare Data Science Center of Taiwan. Using data from the Taiwan Cancer Registry database, we enrolled patients who had pathologically confirmed FIGO stage IIIC endometrial cancer between 2009 and 2015 [15]. The Taiwan Cancer Registry contains prospectively gathered, detailed cancer-related information regarding staging, treatment, and cross-linkage with other population-based registries, including the National Death Registry, to ensure lifelong follow-up. We used the recorded stage; histological type and grade; treatment details such as type of surgery, radiotherapy dose, and technique; and chemotherapy start date in this study. Our study was approved by the institutional review board (IRB: B-EX-108-028, 12 August 2019).

### 2.2. Selection of Study Participants

Female patients with pathologically proven FIGO stage IIIC uterine cancer (ICD-O-3 topography code C54 or C55) were enrolled. We included patients with invasive cancer, and defined endometrial cancer as endometrioid (ICD-O-3 morphological codes 8380, 8381, 8382, and 8383) and non-endometrioid (ICD-O-3 morphological codes 8140, 8310, 8440, 8441, 8460, 8461, and 8480); all other remaining histologies were not included. Study participants were required to undergo at least total hysterectomy, bilateral salpingo-oophorectomy, and lymphadenectomy within 6 months of diagnosis, with pathological staging of the available tumour and nodes. Patients with prior malignancy or distant metastasis were excluded.

### 2.3. Adjuvant Treatment

We defined two patient groups according to the components of the adjuvant treatment: chemotherapy alone or chemoradiotherapy. To exclude patients who experienced significant treatment delays or non-standard therapy, only patients who started chemotherapy within 42 days of surgery were included. Patients who did not receive chemotherapy were also excluded. Patients in the chemoradiotherapy group received external beam radiotherapy within 90 days of surgery to at least 45 Gy and treatment volume denoted as encompassing the pelvis. Patients undergoing brachytherapy without external-beam radiotherapy were excluded.

### 2.4. Patient Covariates and Outcomes Adjuvant Treatment

We extracted the following data from the Taiwan Cancer Registry: patient age (continuous), FIGO IIIC stage (IIIC1 versus IIIC2), AJCC 7th edition staging system (T stage), histological grade and type (endometrioid grade 1–2, endometrioid grade 3, or non-endometrioid), and surgery type (total hysterectomy versus radical hysterectomy), number of LNs removed, and number of pathologically involved LNs. To investigate the value of LN burden-guided treatment, patients were stratified into 1, 2–5, and ≥ 6 LNs according to previous studies [4,16].

The primary outcome investigated was OS, which was defined as the interval from the date of surgery to death from any cause. We linked the Taiwan Cancer Registry with the National Cause of Death Database through a common but anonymised identifier. Patients whose death records were not found were considered alive and censored on the last day of entry into the database (31 December 2017).

### 2.5. Statistical Analysis

Continuous data are presented as median with interquartile range (IQR) or mean ± standard deviation, as applicable, whereas categorical data are presented as numbers (%). Differences between groups were analysed using a chi-square test for categorical variables and an independent *t*-test or Kruskal‒Wallis test for continuous variables.

Survival curves comparing women who received chemoradiotherapy with those who received chemotherapy alone were created using the Kaplan‒Meier method, and differences in survival were assessed using log-rank statistical tests. Cox proportional hazard models were used to evaluate factors associated with OS, and hazard ratios (HRs) and 95% confidence intervals (CIs) were calculated. To account for a guarantee-time bias, we performed a landmark analysis of 1-, 1.5-, and 2-year survivors [17]. Data were analysed using the R software (version 3.6.1, http://www.r-project.org) and SAS (version 9.4, SAS institution Inc., Cary, NC, USA). *p* < 0.05 was considered statistically significant.

## 3. Results

### 3.1. Patient and Tumour Characteristics

Overall, 441 patients who underwent adjuvant chemotherapy alone or chemoradiotherapy for International Federation of Gynaecology and Obstetrics (FIGO) stage IIIC endometrial cancer between 2009 and 2015 were included (Figure 1). Demographic and clinicopathological characteristics according to the treatment group are summarised in Table 1. Of the 441 patients, 142 (32.2%) patients received chemotherapy alone, and 299 (67.8%) received chemoradiotherapy. A higher number of patients with non-endometrioid carcinoma were present in the chemotherapy group than in the chemoradiotherapy group (*p* = 0.04). The FIGO stage, American Joint Committee on Cancer (AJCC) T-stage, surgical type, and number of LNs removed or positive LNs were similar between chemotherapy alone and chemoradiotherapy patients.

### 3.2. Predictors of Survival

The median follow-up period was 3.6 (interquartile range (IQR): 1.8–5.4) years. For the entire cohort, the 5-year OS was 71.1%. The 5-year OS rates of patients with stage IIIC1 and IIIC2 were 74.6% and 66.9%, respectively (*p* = 0.08), whereas the 5-year OS rates of patients divided into three groups based on the number of positive LNs (1, 2–5, and ≥ 6) were 78.3%, 74.8%, and 57.2%, respectively (*p* < 0.001). Patients who received chemoradiotherapy had significantly higher 5-year OS than those who received chemotherapy alone (76.2% versus 61.2%, *p* = 0.002; Figure 2). Similar results were observed in landmark analyses restricted to patients surviving 1, 1.5, and 2 years after surgery (Appendix A).

The results of the univariable and multivariable analyses for all patients are shown in Table 2. In the univariable analysis, AJCC T-stage, histological grade and type, number of positive LNs, and adjuvant chemoradiotherapy were associated with OS. In the multivariable analysis, adjuvant chemoradiotherapy was independently associated with improved OS (hazard ratio (HR): 0.62, 95% confidence interval (CI): 0.43–0.90; *p* = 0.01) compared to chemotherapy alone. The HRs (95% CIs) derived for 2–5 and ≥ 6 positive LNs compared with one LN were 1.56 (0.94–2.59; *p* = 0.09) and 2.22 (1.25–3.95; *p* = 0.006), respectively. Upon excluding patients with missing data for IIIC substage or histological grade, the adjuvant chemoradiotherapy remained independently associated with improved OS (HR: 0.62, 95% CI: 0.41–0.93; *p* = 0.02) (Appendix A).

### 3.3. Influence of Positive LN Number on OS Benefit from Chemoradiotherapy

When stratifying patients based on the number of positive LNs, there was no difference in the OS between chemotherapy alone and chemoradiotherapy for patients with 1 LN (5-year OS: 73.1% versus 80.8%, *p* = 0.31; Figure 3A) or 2–5 positive LNs (5-year OS: 71.4% versus 75.7%, *p* = 0.68; Figure 3B). Patients with ≥ 6 positive LNs showed significant OS benefit associated with chemoradiotherapy compared to those with chemotherapy alone (5-year OS: 70.0% versus 35.9%, *p* < 0.001; Figure 3C). After excluding patients with missing data for IIIC substage or histological grade, chemoradiotherapy remained associated with better 5-year OS than chemotherapy alone in patients with ≥ 6 positive LNs, while patients with 1 or 2–5 positive LNs showed no difference in 5-year OS between chemoradiotherapy and chemotherapy alone (Appendix A).

## 4. Discussion

This is the first study investigating the value of LN number as a predictor for guiding adjuvant treatments in node-positive endometrial cancer patients. We observed an increasing survival benefit from adjuvant chemoradiotherapy as the number of positive LNs increased, beginning at six positive LNs. Our results indicate that nodal number can identify patients whose overall survival may improve from the intensification of adjuvant therapy.

The GOG 258 trial reported that relapse-free survival was not different between adjuvant chemotherapy and chemotherapy alone in locally advanced endometrial cancer (5-year relapse-free survival: 59% versus 58%, *p* = 0.20). However, an exploratory subgroup analysis did not identify a subgroup of patients who might have benefited more from chemoradiotherapy than chemotherapy alone, including patients with FIGO stage IIIC disease [7]. Patients with node-positive endometrial cancer had a high risk for recurrences and the optimal treatment for these patients should be investigated. Our results indicate that chemoradiotherapy provides greater overall survival benefits in patients with ≥ 6 positive LNs. Moreover, chemoradiotherapy can increase treatment-related toxicities and requires longer treatment duration than chemotherapy alone [7,18,19,20]. It is, therefore, essential to weigh the pros and cons of chemoradiotherapy to determine if it is a worthy adjuvant therapy option for the patients. These data indicate the need to be selective of the node-positive endometrial cancer patients treated with adjuvant therapy to achieve optimal outcomes. Hence, stratifying node-positive EC patients based on the number of positive LNs may help determine the optimal adjuvant treatment.

The lymph node ratio (LNR) is a predictor of the tumour burden and the aggressive biological behaviour of the tumour in node-positive endometrial cancers. LNR might be a better predictor of tumour burden than the number of positive LNs [4,21]. LNR is a parameter based on two variables, i.e., the number of positive LNs, and the number of LNs removed. Previous studies used the LNRs of ≤ 10%, 10%–50%, and > 50% to stratify risk groups [4,21]; however, it was unknown if the LNR cut-off values could be used in this study because of the differences in LNs removed. The median number of LNs removed was 11 (range: 1–90) and 20 (range: 1–78) in the study by Chan et al. and Polterauer et al., respectively [4,21]. In this study, we had a higher median number of LNs removed, 29 (IQR: 19–37) in the chemotherapy alone group and 25 (IQR: 18–39) in the chemoradiotherapy group. Due to the vast variability in the number of lymph nodes removed, it was unknown if the LNR cut-off values could be used to guide the choice of adjuvant therapy. For example, if a patient had 29 lymph nodes removed, at least 15 positive lymph nodes would be needed to stratify this patient into the > 50% LNR group. While our study reveals improved survival with adjuvant chemoradiotherapy beginning at six positive LNs, the patient with six positive LNs would be stratified into the 10%–50% LNR group. To avoid heterogeneity in patients due to a broad range of 10%–50% LNR, we used the absolute number of LNs to stratify our patients [4,22]. In addition, it should be noted that there may be a potential therapeutic effect of extensive lymphadenectomy [23]. We found that there was a trend towards improved OS as the number of LNs removed increased. Hence, our findings need to be validated by future studies.

Integration of molecular characteristics may aid in better selecting patients for adjuvant treatment. A recent study investigated the prognostic significance of molecular classification using tissues from the PORTEC-3 trial [24,25,26]. Patients with *POLE*-ultramutated endometrial cancer have excellent clinical outcome regardless of whether they received adjuvant chemoradiotherapy or radiotherapy alone. Patients with p53 mutant disease significantly benefited from adjuvant chemoradiotherapy, while patients with mismatch repair-deficient or no-specific-molecular-profile subgroup had similar survival outcomes with either chemoradiotherapy or radiotherapy alone. Although our study reported that the number of positive LNs could help select patients for adjuvant therapies, incorporating molecular classification may help determine specific subgroups for adjuvant therapies in node-positive endometrial cancer.

This study has several limitations. Despite being based on nationwide registry data, the database may have miscoding of demographic and clinical data. Of particular importance is histological grade, which is a known prognostic factor [13,14,27]; about 12% of patients with endometrioid carcinoma were coded as unknown grade in this study. After excluding these patients, chemoradiotherapy remained independently associated with better OS. Information on lymphovascular space invasion, patterns of recurrences, and other molecular risk factors was unavailable, which could have helped identify subgroups that would be most benefitted from chemoradiotherapy [16,26,28,29,30,31]. Selection bias and residual and unmeasured confounding are potential limitations of this study. The number of patients in our study may not bear the statistical power to determine optimal cut-off values of positive LNs or LNR to guide adjuvant treatment in patients with node-positive endometrial cancer. The imbalance of histology between chemotherapy alone and chemoradiotherapy groups might also affect the analysis and interpretation of the results. Due to the availability of the data, the types of chemotherapy and radiotherapy were not analysed in this study. Doxorubicin plus cisplatin or paclitaxel plus carboplatin was the most commonly used chemotherapy for patients with advanced-stage endometrial cancer in Taiwan [32]. Recently, a randomised trial revealed that the survival outcomes between these chemotherapy types were not different [33]. The number of patients that underwent radiotherapy alone was small in our cohort (*n* = 42); hence, these patients were excluded from this study. Whether number of LNs could be a predictor for radiotherapy alone needs evaluation in future studies. Despite these limitations, a major strength of our study was the use of a nationwide, population-based cohort with prospectively gathered data and cross-linkage with additional population-based registries including the National Death Registry to ensure lifelong follow-up. This study could be a reference from real-world clinical outcome research, which may help select patients with node-positive endometrial cancer who could benefit from chemoradiotherapy in clinical practice.

## 5. Conclusions

Increased number of positive LNs was associated with worse OS in node-positive endometrial cancer patients. Patients with higher number of positive LNs may derive more survival benefit from adjuvant chemoradiotherapy than patients with lower number of positive LNs. These results suggest that the number of positive nodes may be used to identify patients likely to have improved survival outcomes with intensification of adjuvant therapy.

## Figures and Tables

**Figure 1 diagnostics-10-00373-f001:**
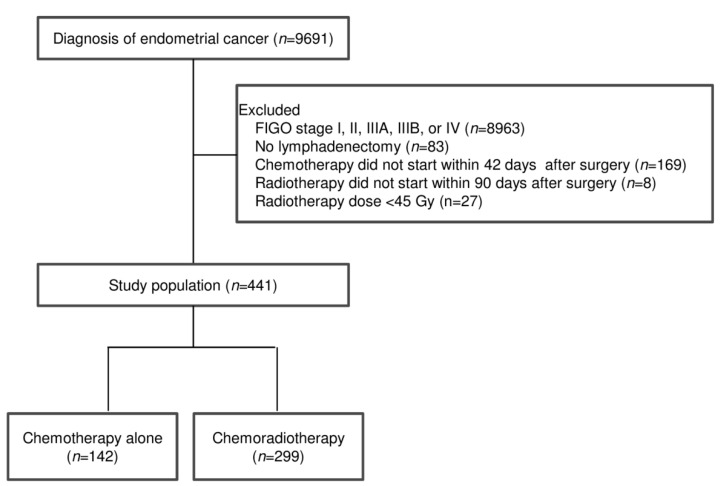
Flow of the patients through the study.

**Figure 2 diagnostics-10-00373-f002:**
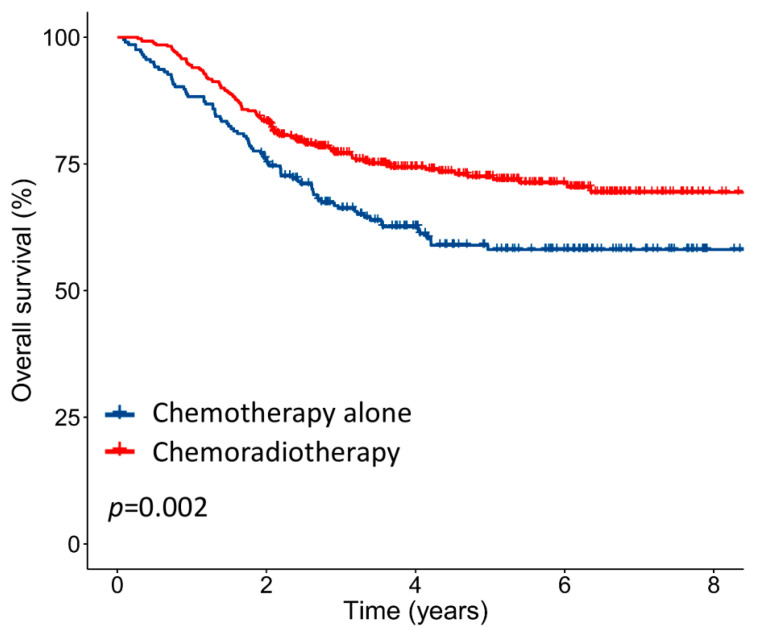
Kaplan–Meier survival curves for the overall survival of all patients stratified by adjuvant treatment type.

**Figure 3 diagnostics-10-00373-f003:**
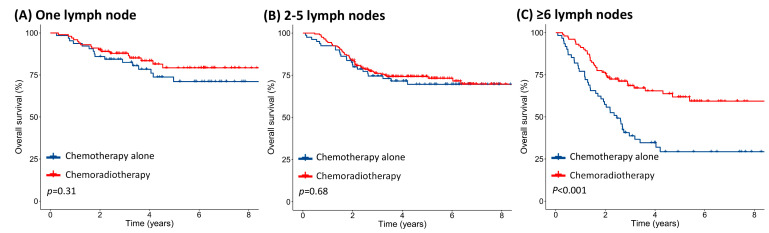
Kaplan–Meier survival curves showing overall survival with chemotherapy alone and chemoradiotherapy considering the number of positive lymph nodes. (**A**) One lymph node, (**B**) 2–5 lymph nodes, and (**C**) ≥ 6 lymph nodes; for adjuvant chemoradiotherapy or chemotherapy alone in all patients.

**Table 1 diagnostics-10-00373-t001:** Patient and tumour characteristics (*n* = 441).

Characteristics	Chemotherapy Alone (*n* = 142)	Chemoradiotherapy (*n* = 299)	*p*-Value
**Age (years)**	57 (51–61)	55 (51–60)	0.23
**FIGO stage**			0.10
IIIC1	83 (58.5)	166 (55.5)	
IIIC2	46 (32.4)	119 (39.8)	
IIIC NOS	13 (9.2)	14 (4.7)	
**AJCC T-stage**			0.78
T1	71 (50.0)	139 (46.5)	
T2	25 (17.6)	58 (19.4)	
T3	46 (32.4)	102 (34.1)	
**Histological grade and type**			0.04
Endometrioid grade 1–2	40 (28.2)	91 (30.4)	
Endometrioid grade 3	48 (33.8)	128 (42.8)	
Endometrioid unknown grade	18 (12.7)	36 (12.0)	
Non-endometrioid	36 (25.4)	44 (14.7)	
**Surgical type**			
TAH/BSO	113 (79.6)	248 (82.9)	0.47
Modified RH	29 (20.4)	51 (17.1)	
**Number of LNs removed**	29 (19–37)	25 (18–39)	0.67
**Number of positive LNs**			
Median (IQR)	3 (1–6)	3 (1–6)	0.45
1	45 (31.7)	77 (25.8)	
2–5	55 (38.7)	144 (48.2)	
≥ 6	42 (29.6)	78 (26.1)	
**Median (IQR) follow-up, years**	3.6 (2.2–6.2)	3.7 (2.3–5.7)	1.00

Abbreviations: FIGO, International Federation of Gynaecology and Obstetrics; IQR, interquartile range; LN, lymph node; NOS, not otherwise specified; RH, radical hysterectomy; TAH/BSO, total abdominal hysterectomy with bilateral salpingo-oophorectomy. Data are median (IQR) or *n* (%).

**Table 2 diagnostics-10-00373-t002:** Univariable and multivariable Cox proportional hazards model for overall survival (*n* = 441).

Variable	Univariable		Multivariable	
	HR (95% CI)	*p*-Value	HR (95% CI)	*p*-Value
**Age, continuous**	1.02 (1.00–1.04)	0.07	1.00 (0.98–1.02)	0.97
**FIGO stage**				
IIIC1	Reference		Reference	
IIIC2 ^a^	1.39 (0.97–1.99)	0.08	1.17 (0.78–1.77)	0.45
**AJCC T-stage**				
T1	Reference		Reference	
T2	1.39 (0.83–2.32)	0.21	1.21 (0.72–2.04)	0.46
T3	2.19 (1.46–3.28)	< 0.001	1.71 (1.11–2.64)	0.01
**Histological grade and type**				
Endometrioid grade 1–2	Reference		Reference	
Endometrioid grade 3 ^b^	1.74 (1.05–2.89)	0.03	1.88 (1.12–3.15)	0.02
Non-endometrioid	4.14 (2.42–7.07)	< 0.001	3.41 (1.93–6.02)	< 0.001
**Surgical type**				
TAH/BSO	Reference		Reference	
Modified RH	1.52 (1.00–2.27)	0.05	1.47 (0.97–2.27)	0.07
**Number of LNs removed**	0.99 (0.98–1.01)	0.28	0.99 (0.97–1.00)	0.06
**Number of positive LNs**				
1	Reference		Reference	
2–5	1.43 (0.87–2.36)	0.16	1.56 (0.94–2.59)	0.09
≥ 6	2.62 (1.58–4.35)	< 0.001	2.22 (1.25–3.95)	0.006
**Adjuvant treatment**				
Chemotherapy alone	Reference		Reference	
Chemoradiotherapy	0.61 (0.42–0.87)	0.007	0.62 (0.43–0.90)	0.01

Abbreviations: CI, confidence interval; HR, hazard ratio. ^a^ Including FIGO stage IIIC2 (*n* = 165) and FIGO IIIC NOS (*n* = 27). ^b^ Including endometrioid grade 3 (*n* = 176) and endometrioid unknown grade (*n* = 54).

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
