# Peer review of "Lymph Node Number Predicts the Efficacy of Adjuvant Chemoradiotherapy in Node-Positive Endometrial Cancer Patients"

_diagnostics, 2020, doi:10.3390/diagnostics10060373_

Round 1

Reviewer 1 Report

Lee and colleagues have written an interesting article on the pediction of effect of adjuvant chemoradiotherapy in node positive endometrial cancer (EC) patients. In large I think the correct types of  analyses have been performed.

I have a few questions to the authors.

  1. Figure 1: fom over 12000 cases with EC 2363 patients were excluded due to non invasive disease. They thus do not have cancer I presume so why are they mentioned? What is the difference between the excluded group 'non FIGO 3c' and 'metastasis at diagnosis'. Only in 83 patients no LND was performed, meanin LND was performed in >99%? In the excluded section it it mentioned 'radioterapy did not start within 100 days after surgery, however in the M&M it is mentioned <90days. this does not coincide. did patients with nodal metastases possibly have other metastases too?
  2. When I look at the study population (n=441) compared to the invasive lesions (n=9691), only 4.5% of patients would seem to have stage FIGO 3c, which is an extremely low % in view of het high LND rate. can this be further detailed?
  3. it is explained in the text the cutoffs are based on two previous articles. However it would have stenghtened the analyses if the cutoffs had been determined data driven rather than on artibrarily. this may have resulted in the same cutffs, but may also have resulted in different and more accurate cutoffs. It would be good to see this analysis performed. similarly the analyses should ideally have been performed using the LNR as well, such as actually hinted at by the authors. Their conclusions might have been strengthened by these additional analyses.
  4. the authors do not mention where the positive nodes have been localised (paraaortally or pelvically) nor whether the localisation of the nodes has any impact on the results. Were all nodal metastases a macrometasetasis?
  5. I cannot find the suppl materials belonging to the manuscript.

Author Response

Lee and colleagues have written an interesting article on the pediction of effect of adjuvant chemoradiotherapy in node positive endometrial cancer (EC) patients. In large I think the correct types of analyses have been performed.

I have a few questions to the authors.

Comment 1: Figure 1: fom over 12000 cases with EC 2363 patients were excluded due to non invasive disease. They thus do not have cancer I presume so why are they mentioned? What is the difference between the excluded group 'non FIGO 3c' and 'metastasis at diagnosis'. Only in 83 patients no LND was performed, meanin LND was performed in >99%? In the excluded section it it mentioned 'radioterapy did not start within 100 days after surgery, however in the M&M it is mentioned <90days. this does not coincide. did patients with nodal metastases possibly have other metastases too?

Response 1: We appreciate the reviewer’s comment. We used the Taiwan Cancer Registry to evaluate treatments in patients with FIGO IIIC endometrial cancer. We concur with the reviewer’s comment that patients with non-invasive disease should not be presented in Figure 1. Patients with FIGO I, II, IIIA, IIIB, and IV were excluded. In addition, lymphadenectomy was routinely performed for patients diagnosed with endometrial cancer in Taiwan during 2009-2015; hence, the rate of lymphadenectomy was high in our population. The adjuvant radiotherapy should start within 90 days after surgery. Patients with nodal metastasis may also have distant metastasis, and these patients should be staged as FIGO stage IVB and thus were excluded. We have corrected Figure 1 according to the reviewer’s valuable comment.

Comment 2: When I look at the study population (n=441) compared to the invasive lesions (n=9691), only 4.5% of patients would seem to have stage FIGO 3c, which is an extremely low % in view of het high LND rate. can this be further detailed?

Response 2: We appreciate the reviewer’s comment. The frequency of FIGO IIIC in our population might be higher than that in the population included in previous studies. Chan et al. enrolled 639 (1.6%) FIGO stage IIIC patients from 40880 patients with endometrial cancer (Br J Cancer 2007, 97, 605-611); Xiang et al. enrolled 5,692 (1.2%) patients with FIGO stage IIIC endometrial cancer from 492,398 patients with endometrial cancer (Gynecol Oncol. 2019 Sep;154(3):487-494). The possible explanation for the higher frequency of FIGO stage IIIC endometrial cancer in our population might be the high rate of lymphadenectomy. Hence, a total of 441 (4.6%) patients with FIGO stage IIIC endometrial cancer from patients with endometrial cancer (n=9691) were enrolled in this study.

Comment 3: it is explained in the text the cutoffs are based on two previous articles. However it would have stenghtened the analyses if the cutoffs had been determined data driven rather than on artibrarily. this may have resulted in the same cutffs, but may also have resulted in different and more accurate cutoffs. It would be good to see this analysis performed. similarly the analyses should ideally have been performed using the LNR as well, such as actually hinted at by the authors. Their conclusions might have been strengthened by these additional analyses.

Response 3: We concur with the reviewer’s valuable comment. The number of patients in our study may not bear the statistical power to determine the cut-off values of positive lymph nodes, which has constituted one of the limitations of this study. The cut-off value we used was derived from a population-based study with 1222 node-positive endometrial cancer patients (Br J Cancer 2007, 97, 605-611). Hence, it might be plausible to use their cut-off in our study. Furthermore, we found that adjuvant chemoradiotherapy improved overall survival in patients with ≥6 positive lymph nodes while the survival benefits were not significant in patients with one or 2-5 positive lymph nodes.

        The lymph node radio (LNR) might be a better predictor of tumor burden than number of positive lymph nodes. Previous studies used the LNRs of ≤10%, 10%-50%, and >50% to stratify risk groups (Br J Cancer 2007, 97, 605-611; Obstet Gynecol 2012, 119, 1210-1218). However, in this study, we had a higher median number of LNs removed than in previous studies: 29 (IQR: 19-37) in the chemotherapy alone group and 25 (IQR: 18-39) in the chemoradiotherapy group. Due to the vast variability in the number of lymph nodes removed, it was unknown if the LNR cut-off values could be used to guide the choice of adjuvant therapy. For example, if a patient had 29 lymph nodes removed, at least 15 positive lymph nodes would be needed to stratify this patient into the >50% LNR group. The optimal cut-off values of LNR and the use of LNR to guide adjuvant treatment are needed to be evaluated in larger studies. To improve this manuscript according to the reviewer’s valuable comment, we revised the Discussion as follows:

In the Discussion section – (Page 7, lines 159-166)

Due to the vast variability in the number of lymph nodes removed, it was unknown if the LNR cut-off values could be used to guide the choice of adjuvant therapy. For example, if a patient had 29 lymph nodes removed, at least 15 positive lymph nodes would be needed to stratify this patient into the >50% LNR group. While our study reveals improved survival with adjuvant CRT beginning at six positive LNs, the patient with 6 positive LNs, would be stratified into the 10%-50% LNR group. To avoid heterogeneity in patients due to a broad range of 10%-50% LNR, we used the absolute number of LNs to stratify our patients [4,19].

In the Discussion section – (Page 8, line 188–190)

The number of patients in our study may not bear the statistical power to determine optimal cut-off values of positive LNs or LNR to guide adjuvant treatment in patients with node-positive endometrial cancer.

Comment 4: the authors do not mention where the positive nodes have been localised (paraaortally or pelvically) nor whether the localisation of the nodes has any impact on the results. Were all nodal metastases a macrometasetasis?

Response 4: We concur with the reviewer’s comment. The FIGO staging system for endometrial cancer has taken location of lymph nodes into account: the FIGO IIIC1 stage indicated the pathologically positive pelvic lymph node disease and FIGO IIIC2 indicated the pathologically positive para-aortic node disease. The number (%) of patients with FIGO IIIC1 and FIGO IIIC2 endometrial cancer are shown in Table 1. Furthermore, in the multivariable Cox proportional hazards model, the FIGO IIIC2 was not associated with worse overall survival compared with FIGO IIIC1 (Table 2).

Comment 5: I cannot find the suppl materials belonging to the manuscript.

Response 5: We appreciate the reviewer’s detailed review and valuable comments. The supplementary materials have been added to the system.

Reviewer 2 Report

the study is interesting and focuses on an open field of investigation.

Nevertheless, there are some issues that should be addressed.

the first one is the imbalance of histologies between the two group. It is not adequately underlined as one of the major limitations of this paper and it is not mentioned in the description of the results (although it is independently associated with a worse OS and it is significant both in uni and multivariate analysis).

If possible I would appreciate a table with types of chemotherapy treatment. 

Also, the type of radiotherapy should be described.

A better description of both PORTEC 3 and GOG 258 is suggested. Moreover, it is not accurate saying that GOG 258 data on OS are not significant because data are not mature (as described in the cited paper Matei NEJM 2019)

Author Response

Reviewer #2:

the study is interesting and focuses on an open field of investigation.

Nevertheless, there are some issues that should be addressed.

Comment 1: the first one is the imbalance of histologies between the two group. It is not adequately underlined as one of the major limitations of this paper and it is not mentioned in the description of the results (although it is independently associated with a worse OS and it is significant both in uni and multivariate analysis).

Response 1: We concur with the reviewer’s comment. To improve the manuscript according to the reviewer’s comment, we have revised the manuscript in the Results and Discussion as follows:

In the Results section – (Page 2, lines 77-79)

A higher number of patients with non-endometrioid carcinoma were present in the chemotherapy group than in the chemoradiotherapy group (p=0.04).

In the Discussion section – (Page 8, line 190–192)

The imbalance of histology between chemotherapy alone and chemoradiotherapy groups might also affect the analysis and interpretation of the results.

Comment 2: If possible I would appreciate a table with types of chemotherapy treatment.

Also, the type of radiotherapy should be described.

Response 2: We concur with the reviewer’s comment. The types of chemotherapy and radiotherapy may affect outcomes in these patients. Further analysis may be needed due to the current availability of these data. To improve the manuscript according to the reviewer’s comment, we revised the Discussion as follows:

In the Discussion section – (Page 8, line 192–196)

Due to the availability of the data, the types of chemotherapy and radiotherapy were not analysed in this study. Doxorubicin plus cisplatin or paclitaxel plus carboplatin was the most commonly used chemotherapy for patients with advanced-stage endometrial cancer in Taiwan [30]. Recently, a randomized trial revealed that the survival outcomes between these chemotherapy types were not different [31].

Comment 3: A better description of both PORTEC 3 and GOG 258 is suggested. Moreover, it is not accurate saying that GOG 258 data on OS are not significant because data are not mature (as described in the cited paper Matei NEJM 2019)

Response 3: We concur with the reviewer’s comment. To improve the manuscript according to the reviewer’s valuable comment, we revised the Introduction section as follows:

In the Introduction section – (Page 2, lines 54-61)

In the PORTEC-3 trial, 170 (25.8%) patients had FIGO stage IIIC endometrial cancer and patients with FIGO stage IIIC disease had high risks of recurrences. The PORTEC-3 trial demonstrated that chemoradiotherapy over radiotherapy alone improves the OS outcome in overall stage III endometrial cancers [6]. In the GOG 258 trial, 538 (73.1%) patients had FIGO stage IIIC endometrial cancer; however, the subgroup analysis showed that these patients might not have benefitted more from chemoradiotherapy than from chemotherapy alone in recurrence-free survival [7]. Hence, there may be need to select patients with FIGO stage IIIC endometrial cancer to receive chemoradiotherapy or chemotherapy alone.

Reviewer 3 Report

Date:2020/5/21

Reviewer's report:

Review of  Lymph node number predicts the efficacy of adjuvant chemoradiotherapy in node-positive endometrial cancer patients

This is an interesting manuscript as it is a multicenter retrospective analysis  of CCRT on this specific group of locally advance endometrial cancer. Thou, there are some limitation in this study such as histological grade, a major prognostic factor, the type, dose, and number cycles  of chemotherapy were not mention too.  Nevertheless, this manuscript will add to growing body of literature on the benefit of concurrent chemotherapy and RT in the treatment of endometrial ca , stage IIIC. However, there are a few issues in the manuscript that need to be addressed prior to publication.

  1. Reason for choosing chemotherapy alone and/or concurrent chemotherapy radiation therapy in this group of patients?

Level of interest: An article of importance in its field

Quality of written English: good

Statistical review: Yes, and I have assessed the statistics in my report.

Declaration of competing interests:

Author Response

Reviewer #3:

Review of Lymph node number predicts the efficacy of adjuvant chemoradiotherapy in node-positive endometrial cancer patients

Comment 1: This is an interesting manuscript as it is a multicenter retrospective analysis of CCRT on this specific group of locally advance endometrial cancer. Thou, there are some limitation in this study such as histological grade, a major prognostic factor, the type, dose, and number cycles of chemotherapy were not mention too. Nevertheless, this manuscript will add to growing body of literature on the benefit of concurrent chemotherapy and RT in the treatment of endometrial ca , stage IIIC. However, there are a few issues in the manuscript that need to be addressed prior to publication.

Response 1: We concur with the reviewer’s comment. This study has several limitations. The histological grade was lacking in about 12% of patients with endometrioid carcinoma. The type and regimens of chemotherapy were not analysed due to availability of the data. To improve the manuscript the reviewer’s comment, we revised the Discussion as follows:

In the Discussion section – (Page 7, line 180 – Page 8, line 196)

This study has several limitations. Despite being based on nationwide registry data, the database may have miscoding of demographic and clinical data. Of particular importance is histological grade, which is a known prognostic factor [13,14,24]; about 12% of patients with endometrioid carcinoma were coded as unknown grade in this study. After excluding these patients, chemoradiotherapy remained independently associated with better OS. Information on lymphovascular space invasion, patterns of recurrences, and other molecular risk factors was unavailable, which could have helped identify subgroups that would be most benefitted from chemoradiotherapy [23,25-29]. Selection bias and residual and unmeasured confounding are potential limitations of this study. The number of patients in our study may not bear the statistical power to determine optimal cut-off values of positive LNs or LNR to guide adjuvant treatment in patients with node-positive endometrial cancer. The imbalance of histology between chemotherapy alone and chemoradiotherapy groups might also affect the analysis and interpretation of the results. Due to the availability of the data, the types of chemotherapy and radiotherapy were not analysed in this study. Doxorubicin plus cisplatin or paclitaxel plus carboplatin was the most commonly used chemotherapy for patients with advanced-stage endometrial cancer in Taiwan [30]. Recently, a randomized trial revealed that the survival outcomes between these chemotherapy types were not different [31].

Comment 2: Reason for choosing chemotherapy alone and/or concurrent chemotherapy radiation therapy in this group of patients?

Response 2: We appreciate the reviewer’s comment. Chemotherapy alone and chemoradiotherapy were chosen because the GOG 258 trial reported that the relapse-free survival were similar between the two treatments in patients with FIGO stage IIIC endometrial cancer. A predictor may help select patients with FIGO stage IIIC to chemoradiotherapy or chemotherapy alone in clinical practice. In addition, our population only had a small number of patients who underwent radiotherapy alone (n=42). Whether number of lymph nodes could be a predictor for radiotherapy alone need evaluating in future studies. PORTEC-3 had also reported improved overall survival in the chemoradiotherapy arm compared to the radiotherapy alone arm for FIGO stage III endometrial cancer. We have revised the manuscript in the Introduction according to the reviewer’s comment as follows:

In the Introduction section – (Page 2, lines 54-61)

The primary treatment for endometrial cancer is surgery, including total hysterectomy, bilateral salpingo-oophorectomy, and lymphadenectomy or sentinel lymph node (LN) mapping. The adjuvant treatments were mainly recommended based on pathological evaluation. However, the optimal adjuvant treatment for patients with node-positive endometrial cancer is controversial [5-9]. The adjuvant chemotherapy or radiotherapy achieved similar overall survival (OS) outcomes; however, their pattern of recurrences differed [10,11]. Combined chemotherapy and radiotherapy might achieve better survival outcomes compared with chemotherapy or radiotherapy alone in patients with node-positive endometrial cancer [11-14]. In the PORTEC-3 trial, 170 (25.8%) patients of them had FIGO stage IIIC endometrial cancer and patients with FIGO stage IIIC disease had high risks of recurrences. The PORTEC-3 trial demonstrated that chemoradiotherapy over radiotherapy alone improves the OS outcome in overall stage III endometrial cancers [6]. In the GOG 258 trial, 538 (73.1%) patients had FIGO stage IIIC endometrial cancer; however, the subgroup analysis showed that these patients might not have benefitted more from chemoradiotherapy than from chemotherapy alone in recurrence-free survival [7]. Hence, there may be need to select patients with FIGO stage IIIC endometrial cancer to receive chemoradiotherapy or chemotherapy alone.

In the Discussion section – (Page 8, lines 196-199)

The number of patients that underwent radiotherapy alone was small in our cohort (n=42); hence, these patients were excluded from this study. Whether number of LNs could be a predictor for radiotherapy alone need evaluating in future studies.

Round 2

Reviewer 1 Report

The authors have tried to answer all questions in a satisfactory manner.

Reviewer 2 Report

In my opinion, the authors might review the conclusions highlighting also the limits of the article. Nevertheless, I think it is acceptable also in the present form